# The Detection and Classification of Scaphoid Fractures in Radiograph by Using a Convolutional Neural Network

**DOI:** 10.3390/diagnostics14212425

**Published:** 2024-10-30

**Authors:** Tai-Hua Yang, Yung-Nien Sun, Rong-Shiang Li, Ming-Huwi Horng

**Affiliations:** 1Department of Biomedical Engineering, National Cheng Kung University, Tainan 701, Taiwan; 2Department of Orthopedic Surgery, College of Medicine, National Cheng Kung University Hospital, National Cheng Kung University, Tainan 704, Taiwan; 3Department of Computer Science and Information Engineering, National Cheng Kung University, Tainan 701, Taiwan

**Keywords:** medical image computer-aided diagnosis system, scaphoid bone, scaphoid fractures, multi-view detection and segmentation, convolutional neural network

## Abstract

Objective: Scaphoid fractures, particularly occult and non-displaced fractures, are difficult to detect using traditional X-ray methods because of their subtle appearance and variability in bone density. This study proposes a two-stage CNN approach to detect and classify scaphoid fractures using anterior–posterior (AP) and lateral (LA) X-ray views for more accurate diagnosis. Methods: This study emphasizes the use of multi-view X-ray images (AP and LA views) to improve fracture detection and classification. The multi-view fusion module helps integrate information from both views to enhance detection accuracy, particularly for occult fractures that may not be visible in a single view. The proposed method includes two stages, which are stage 1: detect the scaphoid bone using Faster RCNN and a Feature Pyramid Network (FPN) for region proposal and small object detection. The detection accuracy for scaphoid localization is 100%, with Intersection over Union (IoU) scores of 0.8662 for AP views and 0.8478 for LA views. And stage 2: perform fracture classification using a ResNet backbone and FPN combined with a multi-view fusion module to combine features from both AP and LA views. This stage achieves a classification accuracy of 89.94%, recall of 87.33%, and precision of 90.36%. Results: The proposed model performs well in both scaphoid bone detection and fracture classification. The multi-view fusion approach significantly improves recall and accuracy in detecting fractures compared to single-view approaches. In scaphoid detection, both AP and LA views achieved 100% detection accuracy. In fracture detection, using multi-view fusion, the accuracy for AP views reached 87.16%, and for LA views, it reached 83.83%. Conclusions: The multi-view fusion model effectively improves the detection of scaphoid fractures, particularly in cases of occult and non-displaced fractures. The model provides a reliable, automated approach to assist clinicians in detecting and diagnosing scaphoid fractures more efficiently.

## 1. Introduction

The human wrist consists of eight carpal bones, with the scaphoid bone being the largest. Located under the thumb and near the arm, as indicated by the red box in Figure 1, the scaphoid is prone to injury because of its location. It can fracture not only from severe accidents but also from falls where one braces themselves with the wrist or during athletic activities. A scaphoid fracture may not always cause severe pain; symptoms can be mild, leading individuals to believe they have merely sprained their wrist and that it will heal without medical intervention. Scaphoid fractures account for 2% to 7% of all fractures and 60–70% of carpal bone fractures. General treatment often involves screw fixation surgery because of its short operation time and good recovery outcomes. However, accurate localization of both the scaphoid and the fracture site is essential for implanting the screw at the optimal angle. Physicians typically use X-rays for diagnosis, but since X-rays are two-dimensional projections of a three-dimensional object, the scaphoid bone may overlap and not be fully visible in the two-dimensional image. This is particularly problematic for non-displaced fractures, which can be challenging to detect with the naked eye and are termed occult fractures, as shown in Figure 2.

Brian Gibney estimated the occurrence rate of occult fractures to be approximately 7–21% [1]. In addition to occult fractures, varying bone density among patients can complicate diagnosis. To address this, doctors often use both the anterior–posterior (AP) view and lateral (LA) view for diagnostic purposes. The AP view shows the scaphoid bone’s front and back, while the LA view provides side information. Combining these views can offer a more comprehensive basis for assessment. If discomfort persists after treatment, further observation with tools like Computed Tomography (CT) or Magnetic Resonance Imaging (MRI) may be used for a more detailed diagnosis or surgical planning. However, such equipment is very expensive, highlighting the importance of improving diagnostic accuracy with X-ray images.

Recent advancements in convolutional neural networks (CNNs) have led to significant success in detection and classification tasks within computer vision. Various models have been proposed, including one-stage detection models like YOLO [2], RetinaNet [3], and FCOS [4], as well as two-stage detection models such as Faster RCNN [5] and Cascade RCNN [6], and classification models like ResNet [7], DenseNet [8], and EfficientNet [9]. These models are widely used because of their ability to learn automatically and efficiently, offering advantages over traditional image processing methods. This study utilizes a deep learning object detection algorithm to develop a computer-assisted diagnostic system for scaphoid fractures. The system includes the following components: first, it locates the scaphoid bone in X-ray images of the wrist from the anterior–posterior view, and second, it detects fracture areas in both anterior–posterior and lateral views.

Object detection algorithms focus on extracting important regions for further analysis. One-stage detectors can quickly produce results but may sacrifice some precision, while two-stage detectors offer higher precision at the cost of increased computation time. One-stage methods predict object classes and deviations from dense anchor boxes or points in a single step, which is fast and less hardware-intensive because of fewer parameters. One-stage methods include anchor-based approaches like SSD [10], YOLO [2], and Foveabox [11]. These methods face challenges due to the imbalance between positive and negative samples, leading to the development of loss functions such as Focal Loss [3] to address this issue. Despite their lower precision compared with two-stage methods, their speed makes them suitable for certain tasks. Two-stage methods, such as those in the RCNN family (e.g., RCNN [12], Faster RCNN [5], and Mask RCNN [13]), first propose regions of interest using a Region Proposal Network (RPN) before performing more precise detection in the second stage. These methods provide higher detection accuracy due to the initial selection of candidate areas but require more computational resources.

In response to the success of CNNs in computer vision, computer-assisted diagnostic systems for fracture detection have emerged. These systems can be categorized into two main types as follows: those that simply classify the presence of a fracture and those that use additional information for more accurate detection or classification. For instance, Yadav et al. used a feature extractor followed by a fully connected layer to classify fracture images [14]. Raisuddin et al. [15] used landmark detection and SE ResNet50 [16] for feature extraction and classification. Alfred Yoon employed two Efficient-B3 models to re-classify images flagged as normal by the first model [17]. These methods, however, often fall short of pinpointing the exact fracture location, relying on gradient-weighted class activation maps [18], which can be imprecise.

Other approaches incorporate additional information for fracture detection. Lindsey et al. used an encoding–decoding architecture to locate fractures by generating a mask of the fracture area [19]. Chen et al. leveraged symmetry by flipping pelvic images and using a Siamese encoding structure to create a heatmap of fracture locations [20]. Lee et al. combined X-rays with semantic information to classify fractures [21]. These methods highlight the need for multi-source information to improve detection accuracy but face challenges such as sample imbalance and variability in image quality and annotations.

In previous research, we developed a method for scaphoid fracture detection from anterior–posterior views, achieving classification results with 82.9% accuracy, 73.50% recall, and 89.8% precision, and detection results with 85.3% accuracy, 78.9% recall, and 89.4% precision [22]. Although precision was high, there was room for improvement in accuracy and recall. This study aims to enhance these metrics by incorporating lateral views into a multi-view fusion approach for improved classification and detection. Lee et al. [23] used an artificial intelligence model to detect the following common wrist fractures: distal radius, ulnar styloid process, and scaphoid. The resulting area under the curves (AUCs) for detecting distal radius, ulnar styloid, and scaphoid fractures per wrist were 0.903 (95% C.I. 0.887–0.918), 0.925 (95% C.I. 0.911–0.939), and 0.808 (95% C.I. 0.748–0.967), respectively. The work by Singh et al. [24] used 525 X-ray images, 250 of which were normal scaphoids, 219 were fractured scaphoids, and 56 were occult fracture X-rays, to classify two disease classes. The proposed model achieved sensitivity, specificity, accuracy, and AUC of 92%, 88%, 90%, and 0.95, respectively.

Several studies on multi-view classification have shown that combining multiple views can enhance model performance [25,26,27]. For example, Peng et al. used frontal, lateral, and posterior–anterior chest X-ray images combined with different CNN models for pneumonia classification [28]. Hassan Nasir Khan et al. utilized four views of breast X-ray images and merged features for breast cancer diagnosis [29]. Yanhan Li et al. employed a Siamese network to combine lung ultrasound images with biomedical indicators for COVID-19 detection [30].

In multi-view detection tasks, Yue Wang et al. encoded images from various views using ResNet and FPN, converting them from 2D to 3D space for refined object queries [31]. Guojun Wang combined color and point cloud images from different views, using attention mechanisms to fuse features and predict 3D bounding boxes [32].

Clinical systems often use multiple X-ray views to diagnose scaphoid fractures, combining frontal, lateral, and anterior–posterior views to simulate the clinical diagnostic process. This study develops the following models: the first uses Faster RCNN for scaphoid bone localization, reducing computation costs, and the second identifies fracture regions from multiple views, combining features to enhance classification performance.

The main contributions of this article are as follows:A clinically inspired model utilizing multi-view X-ray information is developed for scaphoid fracture classification and localization, improving diagnostic efficiency.The multi-view fusion method enables 2D space detection without 3D transformation, facilitating training with cost-effective X-ray images and addressing challenges in single-view detection.The proposed method achieves high classification performance with 89.94% accuracy, 87.33% recall, and 90.36% precision, with improved recall rates in both frontal and lateral view detectors due to multi-view fusion.

## 2. Database and Methods

### 2.1. Database

In this study, all experimental X-ray images were collected from the National Cheng Kung University Hospital (NCKUH) in Taiwan, including 175 sets of anterior–posterior and lateral X-ray images of the arm of adults. Specifically, 75 fractured instances of surgical verification were selected as the positive images, and 100 normal instances were considered the negative images. These images were annotated by two clinical physicians from the Department of Orthopedics. The annotations included identifying the presence of fractures and marking their locations. Fractures marked in the annotations were confirmed through surgery, and their locations were noted using the annotation program developed for this study. Both anterior–posterior and lateral views were annotated, with annotations being angle-specific. This approach was necessary because fractures or cracks have specific directions, and using a horizontal bounding box might cover non-fracture areas as well. Consequently, directed bounding boxes were used to mark fracture locations accurately. Annotating both views allowed for cross-referencing, enabling doctors to highlight previously inconspicuous areas. This process, simulated during model training, enhanced the detection of occult and non-displaced fractures.

### 2.2. Methods

The main objective of this article is to locate the scaphoid bone in X-ray images and assist doctors in determining whether it is fractured, as well as in identifying the fracture area. Displaced fractures are typically easy to detect in X-ray images, allowing doctors to diagnose them quickly. In contrast, non-displaced and occult fractures are more challenging to identify because of X-ray projection shadowing and variations in bone density among patients. Additionally, discrepancies in interpretation between radiologists and orthopedic surgeons are common. For fractures not visible in the anterior–posterior view, doctors must examine the scaphoid bone for discontinuities, making the diagnosis time-consuming and demanding a thorough understanding of the scaphoid.

Our model is designed to improve upon traditional diagnostic methods by integrating both anterior–posterior and lateral views of X-ray images. This approach provides additional side information often overlooked in standard assessments. The first stage of the model uses Faster RCNN to extract the scaphoid bone from both views of the arm X-ray image. These extracted regions are then processed in the second stage using a multi-view fusion model developed in this study. The model separately analyzes fracture areas from both views and fuses the features to enhance discriminative capability. The fused features are used to assess whether the scaphoid bone is fractured, and the classification scores help refine the detection of fracture areas for greater accuracy. An overview of the system is illustrated in Figure 3.

Figure 3 provides a schematic of the system’s workflow. Initially, wrist X-ray images from two perspectives are collected. These images are processed to detect the scaphoid bone region, a critical component of the wrist. Once the scaphoid region is identified, the multi-view fusion model evaluates images from both perspectives to classify and locate fractures. This fusion approach aims to enhance the accuracy and reliability of fracture detection.

#### 2.2.1. Scaphoid Bone Area Detection

As mentioned earlier, the first stage aims to extract the scaphoid bone from the entire wrist X-ray image. For this purpose, we used a Faster RCNN model combined with a Feature Pyramid Network (FPN). Faster RCNN is a renowned two-stage detector that incorporates a Region Proposal Network (RPN), which enhances the speed of the detector by replacing the traditional algorithmic approach used in the initial stage. The Feature Pyramid Network improves the utilization of feature maps at various scales, addressing the issue that deeper networks might lose small objects.

In our research, if the first stage fails to locate the scaphoid bone accurately, subsequent fracture detection cannot proceed effectively. Therefore, we employed the two-stage Faster RCNN model with the FPN at this stage, despite its slightly slower processing time, because of its improved accuracy. Unlike the original Faster RCNN model, which uses a single-level feature map to generate candidate boxes, the FPN integrates feature maps from multiple levels. This approach helps retain small objects that might otherwise be lost during pooling, thereby enhancing detection performance for small and medium-sized objects.

Considering that a typical wrist X-ray image measures approximately 2000 × 1600 pixels and the scaphoid bone region is around 150 × 150 pixels, the scaphoid falls into the category of small and medium-sized objects. To address this, we used the Feature Pyramid Network (FPN), which upsamples and enlarges deep features to match the size of previous layers and then integrates them using a convolutional layer. Specifically, after incorporating the FPN, a 1 × 1 convolutional layer adjusts the channel number to 256, and deep features are upsampled and combined with the previous layer’s features. A final 3 × 3 convolutional layer refines the feature map. This multi-layer approach, with varied anchor box sizes and aspect ratios, enhances overall detection performance.

Additionally, the Faster RCNN model in this study uses RoI Align to generate feature maps of consistent dimensions for different proposals. RoI Align addresses the limitations of RoI Pooling by avoiding rounding issues associated with dividing regions into smaller blocks. Instead, RoI Align performs bilinear interpolation on these blocks, yielding more accurate feature maps and improving detection results.

To handle multiple bounding boxes that may overlap, Non-Maximum Suppression (NMS) is applied to retain the most accurate boxes. NMS starts by selecting the bounding box with the highest score and storing the remaining boxes in an output list. It then calculates the Intersection over Union (IoU) between the selected box and the remaining boxes. If the IoU exceeds a predefined threshold, the overlapping box is removed from the list. This process repeats until only the boxes with the highest scores remain, effectively representing the detected objects.

#### 2.2.2. Fracture Classification and Detection

The following sections will detail the multi-view fusion model proposed in this article. Multi-view image techniques have shown promise in enhancing classification capabilities, particularly in clinical settings where accurately locating fractures is crucial. The effective fusion of multi-view data is essential to improving fracture detection beyond what can be achieved with 2D projection images alone. The model presented consists of the following key components: the backbone network, the context extractor, the Feature Pyramid Network (FPN), and the multi-view fusion module, as illustrated in Figure 4.

Backbone Network: The anterior–posterior (AP) and lateral (LA) views are first processed through the backbone network to extract features. The backbone network must have a sufficiently large receptive field to capture meaningful features. Deeper networks like ResNet, which use residual connections to mitigate issues like gradient vanishing, are ideal. ResNet-152 was found to be the most effective in our experiments among ResNet-50, ResNet-101, and ResNet-152. It provides deep, high-quality features, with the last three blocks of ResNet-152 feeding into the context extractors.

Context Extractor: To address the challenge of detecting occult or non-displaced fractures, which can be camouflaged by lighting or bone density variations, the context extractor expands the receptive field without pooling, using dilated convolutions. This approach, as demonstrated by Mei et al. [33], helps to extract rich semantic features across a broader context. 

Each context extractor contains four context extraction branches, as shown in Figure 5. First, the 1×1 convolution compresses the number of channels to 1/4 of the original and then uses the $k_{i}\times k_{i}$ convolution and the  3×3 dilated convolution with a dilation rate of $r_{i}$ to extract context-aware features, where i represents the context extraction branch of $i^{th};$ $k_{i}\in \left[1,3,5,7\right],r_{i}\in \left[1,2,4,8\right]$. Each convolutional layer is followed by a set of batch normalization layers and a ReLU. The output of $i^{th}$ is added with the output of ${\left(i+1\right)}^{th}$ to stack the different ranges of receptive fields together. Finally, the outputs of the four context extraction branches are concatenated together and use 1×1 convolution to integrate all features. Such a design enables the context extractor to obtain a wide range of rich perceptual features, allowing the model to have contextual reasoning capabilities and to see finer differences in feature maps. The FPN enhances the feature maps extracted from different scales by combining deep and shallow features, improving the detection of small- and medium-sized objects like the scaphoid bone.

Multi-View Fusion Module: After feature extraction and context enhancement, the FPN outputs from the AP and LA views are fed into the multi-view fusion module. This module integrates features from both views to enhance fracture classification and detection accuracy. The multi-view fusion module uses the FPN features from the AP views and LA views for fusion, and the feature maps of the corresponding layers are input into the fusion block, as shown in Figure 6.

The fusion block first performs global average pooling on the feature maps of the two views and outputs [B, C, 1, 1] to flatten the feature maps and concatenate them. Then, the designed weight transform calculates the common important information of the two views and then multiplies this weight back to the respective views. Finally, the features of the two views are concatenated together for the final classification. The goal is to find as much of the fractured scaphoid region as possible. We added the output of $i^{th}$ to the output of ${\left(i+1\right)}^{th}$, where i represents the *i*th fusion block.

The algorithm is as follows:(1)Feature Extraction: Both AP and LA views are processed through the backbone network to extract initial features.(2)Contextual Information: The context extractor refines these features by expanding the receptive field and integrating contextual information.(3)Feature Fusion: The FPN combines features from multiple scales to improve detection performance.(4)Final Classification and Detection: The multi-view fusion module combines features from both views to classify fractures and re-evaluate detection boxes for precise localization.

The model aims to improve the accuracy of fracture detection, particularly for occult and non-displaced fractures, by leveraging comprehensive multi-view information and advanced feature extraction techniques.

Most of the detection methods, regardless of whether they are one-stage or two-stage methods, only use a single scale for prediction. However, in order to extract better features, the depth of the model is often very deep to prevent the loss of information on small objects. FPN can combine deep and shallow information well, allowing the feature map of each scale to be better used and greatly improving the detection results of small objects. The ground truth of the fractures in this article also belongs to small objects, so an FPN is added after the context extractor. The structure here is the same as that of Faster RCNN. First, the 1×1 convolution layer is used to adjust the number of channels of the input feature map to 256; then, the $i^{th}$ feature map is upsampled and added with ${\left(i-1\right)}^{th}$ feature maps; and finally, the information is integrated through a 3×3 convolution.

Weight transform (see Figure 7) is mainly used to combine information from the AP and LA views. It is too rough to classify the features directly after merging, as mentioned above. In general, the importance of the AP view is higher than that of the LA view. Therefore, classification after direct concatenation may cause all the weights to be pulled away by the AP views and results in the loss of the features of the LA views. Therefore, we design a weight transformation, which combines the features of the AP views and the LA views, calculates the common important information of the two views, and then multiplies them back to their respective views. In this way, the information of the LA views is preserved, preventing its weight from being completely discarded. Equations (1) and (2) show the module construction by different feature maps AP*_i_* and LP*_i_* of scale *i*.
(1)$$MultiviewFusionModule=\sum_{i=0}^{l}C\left(AP_{i}\times W_{i},LA_{i}\times W_{i}\right)$$
(2)$$W_{i}=Sigmoid\left(L\left(ReLU\left(L\left(C\left(GAP\left(AP_{i}\right),GAP\left(LA_{i}\right)\right)\right)\right)\right)\right)$$
where l represents the number of layers of the feature pyramid; L represents the fully connection layer; and C represents the concatenation. The number of channels output by the two fully connected layers is 1/2 of the number of input channels.

The architecture of the detection branch in the second stage is a fully convolutional network. The classification of the detection branch and the prediction head of the detection is the same and consists of 3×3 convolutional layers, a batch normalization layer, ReLU, and 3×3 convolutional layers. The number of output channels of the classification branch is A×C, and the number of channels of the detection branch is  A×5. A and C represent the number of anchor boxes and categories, respectively, and 5 represents $\left(x,y,w,h,\theta \right)$, representing the center point coordinates, width, high, and angle of the prediction box [22]. Each layer of the FPN has its prediction head, and finally, the results of each layer are merged to unify match with the ground truth and calculate the loss. The multi-view fusion classification branch consists of a fully connected layer, ReLU, and a fully connected layer. The first fully connected layer inputs the refined features by the multi-view fusion module, and the output channel is half of the number of inputs. The output of the second fully connection layer is 2, indicating the probability of fracture and non-fracture.

To pass information to fracture detection in two different views through this classification branch, the classification results are rescored for the prediction boxes in the final prediction stage. In the most perfect case, both the classification score and the predicted box score will be close to 1, so when both branch predictions are at their highest, the sum is 2 and the smallest is 0. We took 0.5 as the threshold of the classification branch to keep the classification score greater than or equal to the threshold and set the classification score to 0 if it was smaller than the threshold. The purpose of this is to give priority to the classification results. Because the classification branch directly uses the multi-view feature, it can better distinguish whether the scaphoid is fractured or not. The rescoring formula is shown in Equation (3).
(3)$$Rescore=Threshold\left(Score_{cls},0.5\right)+Score_{det}$$

The model in the second stage consists of three parts, namely, the detector for AP views, the detector for LA views, and the classifier for multi-view fusion. The following describes their loss functions. The detector part is similar to Faster RCNN but regresses an additional angle. First, the ground truth box $\left(x,y,w,y,\theta \right)\;$ and prediction box $\left(x^{\prime },y^{\prime },w^{\prime },h^{\prime },\theta^{\prime }\right)$ are encoded as $v=\left(t_{x},t_{y},t_{w},t_{h},t_{\theta }\right)$ and $v^{\prime }=\left(t_{x}^{\prime },t_{t}^{\prime },t_{w}^{\prime },t_{h}^{\prime },t_{\theta }^{\prime }\right)$, and the anchor box is expressed as $\left(x_{a},y_{a},w_{a},h_{a},\theta_{a}\right)$. v and $v^{\prime }$ are defined in Equations (4) and (5).
(4)$$t_{x}=\frac{x-x_{a}}{w_{a}},t_{y}=\frac{y-y_{a}}{h_{a}}\;t_{w}=log\left(\frac{w}{w_{a}}\right),t_{h}=log\left(\frac{h}{h_{a}}\right)\;t_{\theta }=\frac{4\theta }{\pi }$$
(5)$$t_{x}^{\prime }=\frac{x^{\prime }-x_{a}}{w_{a}},t_{y}^{\prime }=\frac{y^{\prime }-y_{a}}{h_{a}}\;t_{w}^{\prime }=log\left(\frac{w^{\prime }}{w_{a}}\right),t_{h}^{\prime }=log\left(\frac{h^{\prime }}{h_{a}}\right)\;t_{\theta }^{\prime }=tanh\left(\theta^{\prime }\right)$$

Then, SmoothL1 loss is used to regress the positive samples, where SmoothL1 loss is defined in Equation (6).
(6)$$L_{reg}=\frac{1}{N_{pos}}\sum_{1}^{N_{pos}}Smooth_{L1}\left(v_{i}^{\prime }-v_{i}\right)$$

The classification prediction head of the detection branch uses the Binary Cross Entropy loss, defined in Equations (7) and (8) as follows: (7)$$L_{cls}=\frac{1}{N_{pos}}\sum_{i=1}^{N}BCE\left(y_{i},p_{i}\right)$$
(8)$$BCE\left(y,p\right)=\left\{\begin{array}{c} -log\left(p\right)ify=1\\ log\left(1-p\right)ify=0 \end{array}\right.$$

Since the number of negative samples is much larger than that of positive samples, we divide by the number of positive samples rather than the total number of samples to take the average, which can prevent the classifier from being affected by too many negative samples and can better solve the problem of sample imbalance. Where N represents the number of total samples, and $N_{pos}$ represents the number of positive samples. The loss function of multi-view fusion classification is the same as that of Faster RCNN, and both use the Cross Entropy loss. The total loss is defined in Equation (9).
(9)$$L=L_{AP_{cls}}+L_{AP_{reg}}+L_{LA_{cls}}+L_{LA_{reg}}+L_{multi-view\_cls}$$

### 2.3. Training and Testing Database

This study was conducted in collaboration with National Cheng Kung University Hospital and utilized a dataset of 175 wrist X-ray images, including both anterior–posterior (AP) and lateral (LA) views. The labels for these images included information on whether a fracture was present and the specific location of the fracture. All fractures identified in the dataset were confirmed through surgery, and their locations were accurately marked. The provided labeling program allows simultaneous annotation of both AP and LA views. Given that fractures can be oriented, traditional horizontal bounding boxes might cover non-fracture regions. To address this, oriented bounding boxes were used to label fractures. This approach allowed for more precise annotation, as it better accommodates the orientation of fractures. Simultaneous labeling from two different views offers the advantage of cross-referencing, enabling doctors to annotate difficult-to-detect areas more effectively. This comprehensive labeling enhanced the model’s ability to identify subtle differences, improving its detection of occult and non-displaced fractures.

The performance of the proposed method was assessed using five-fold cross-validation. In this approach, the dataset was divided into five subsets. Each subset served as a testing set while the remaining four subsets were used for training. Specifically, the training dataset consisted of approximately 140 images, including 60 images of fractures and 80 normal images, while the testing dataset comprised about 35 images, including 15 fracture images and 20 normal images, as illustrated in Table 1. This cross-validation technique ensured that all data were utilized for both training and testing, helping to mitigate potential biases introduced by random data distribution. It provided a robust evaluation of the proposed method and model by ensuring that each data point was used in both training and testing phases, thereby improving the reliability of the performance metrics.

### 2.4. Metrics

The system setup included Ubuntu 20.4, an Intel(R) Core(TM) i7-10700k CPU @ 3.80 GHz, and an NVIDIA RTX 3090 graphics card. The deep learning framework employed was PyTorch 1.9.0. For scaphoid bone detection, the images were resized to 1400 × 1200 pixels, considering the aspect ratio and the average dimensions of arm X-ray images. Contrast Limited Adaptive Histogram Equalization (CLAHE) was used to improve contrast and highlight image details. The training involved 5000 iterations with a batch size of 1. The initial learning rate was 0.001, and Stochastic Gradient Descent (SGD) was the chosen optimizer, with a weight decay of 0.0001 and a momentum of 0.9. Data augmentation included random horizontal flipping and random contrast adjustment. Additionally, to accommodate varying contrast levels observed in the images provided by doctors, random contrast adjustment was used during training to enhance adaptability.

In the second stage of fracture detection and classification, the images were resized to 256 × 256 pixels, and CLAHE was applied again for contrast adjustment. The entire training process consisted of 1500 iterations with a batch size of 12. The initial learning rate was set to 0.0001, and the Adam optimizer was used with a weight decay of 0.0001. The training process also included random horizontal flipping and random contrast adjustment for data augmentation. The designed anchor boxes had aspect ratios of 1, 1.15, 2, 2.25, and 3, and scales of 1, 21/5, 22/5, 23/5, and 24/5, with sizes of 9 × 9, 4 × 4, and 1 × 1 pixels.

In the classification task, we evaluated the results using a confusion matrix, which is a matrix with the x-axis representing the ground truth and the y-axis representing the model predictions. The matrix is divided into true positive (TP), false positive (FP), true negative (TN), and false negative (FN) categories based on the actual and predicted outcomes. The confusion matrix is shown in Figure 8. We used traditional evaluation metrics to assess the results, which include accuracy, recall, and precision. These metrics are defined as follows:(10)$$Accuracy=\frac{TP+FN}{TP+FP+TN+FN}$$
(11)$$Recall=\frac{TP}{TP+FN}$$
(12)$$Precision=\frac{TP}{TP+FP}$$
(13)$$IoU=\frac{A\cap B}{A\cup B}$$

Accuracy represents the overall proportion of correct predictions. Recall (also known as sensitivity or the true positive rate) is the ratio of patients correctly predicted by the model to the total number of actual patients. Precision is the ratio of patients correctly predicted by the model to all instances predicted as patients. Specificity (or the true negative rate) indicates the proportion of non-patients correctly identified by the model. IoU (Intersection over Union) measures the overlap ratio between the ground-truth box and the predicted box.

In the first stage of scaphoid detection, we performed the evaluation using accuracy, recall, precision, and IoU. In the second stage, fracture detection was assessed using the same four metrics as in the first stage. For classification, we used the following five metrics: accuracy, recall, precision, sensitivity, and specificity.

## 3. Results

Table 2 shows the results of scaphoid detection, indicating that Faster RCNN outperforms YOLOv4. Although YOLOv4, a one-stage method, also yields good results with a recall of 97.50% and an accuracy of 100.0%, the two-stage Faster RCNN method achieves superior performance in the scaphoid bone detection task, with both recall and precision at 100%.

Ablation experiments were conducted to evaluate the model proposed in this study, focusing on single AP view versus multi-view comparisons, the utility of the context extractor, and the effectiveness of the rescoring strategy. The experimental procedure was incrementally stacked in three stages as follows: (1) Single-View and Multi-View Comparisons: The initial experiment did not include context extractors or rescoring strategies. (2) Context Extractor Utility: The second experiment compared the utility of the context extractor in a multi-view setup. This experiment showed improved performance compared with the first but did not include the rescoring strategy. (3) Rescoring Strategy: The third experiment evaluated the rescoring strategy in conjunction with the multi-view context extractor, demonstrating better performance than the second experiment.

As noted earlier, relying solely on AP views for fracture detection and classification overlooks lateral information, which can make it challenging to detect certain fractures in X-ray images, such as non-displaced or occult fractures. In this subsection, we validate the second-stage fracture detection and classification by comparing single-view and multi-view approaches. The model structure for a single AP view is similar to that of the multi-view, with the primary difference being whether the classification branch utilizes the multi-view fusion module. The single AP view model and LA view model use global average pooling (GAP) to flatten and combine features from different layers via Feature Pyramid Networks (FPNs) [22]. The multi-view model uses the proposed method in this article. The results of classification and detection branches are shown in Table 3 and Table 4. 

Although the accuracy of the classification branch did not change significantly, precision decreased by 12.25%, while recall increased by 14.41%, as shown in Table 2. Table 3 shows the fracture detection results obtained by using the AP view, LA view, and multi-view. Although the precision of multi-view methods was reduced by 7.15%, the accuracy and recall improved by 4.44% and 17.75%, respectively. This suggests that using multi-view approaches results in more balanced recall and precision, indicating overall better performance. When using a single AP view, the model tends to classify most scaphoid cases as normal, resulting in higher precision but lower recall. In contrast, the multi-view approach achieves better accuracy and significantly improved recall, allowing the model to better identify less apparent scaphoid fractures. These results indicate that the model is better at distinguishing non-displaced or occult fractures. For the LA view, the fracture detection branch shows improvements across all three metrics as follows: accuracy increased by 4.56%, recall by 0.7%, and precision by 3.15%. As demonstrated by the results in the tables, the multi-view approach consistently outperforms the single AP view, whether for classification or fracture detection.

As mentioned earlier, doctors diagnose fractures by examining the continuity in the scaphoid edge. This requires comprehensive information about the entire scaphoid bone. To address this, we introduced context extractors to help the model capture more detailed information. Table 5 displays the performance of the context extractor. The experimental results indicate that adding the context extractor improves the performance of each model branch, as it allows for the extraction of more complete global features through dilated convolutions of varying sizes. This enhancement helps the model better account for variations in lighting, shadows, or bone densities.

Table 6 displays the results of scaphoid area detection in the AP and LA views during the first stage obtained by using Faster RCNN. The table shows that scaphoid detection achieves 100% accuracy in both views, with a high enough IoU even when the LA view is occluded. This demonstrates that the two-stage detector performs well, and even small objects can be detected effectively when combined with the FPN.

Table 7 and Table 8 present the results of the final five-fold cross-validation for the second-stage fracture classification using both the original single AP view and the multi-view fusion model based on our proposed method of classification and detection branches. The accuracy of the classification branch increased by 2.25%, and the recall rate improved significantly by 12.07%, as shown in Table 6. While the increase in accuracy is modest, the model now correctly identifies a larger number of fractured scaphoid bones that were previously classified as normal, leading to a substantial improvement in recall.

Table 8 indicates that the accuracy of the two views improved by 6.16% and 14.07%, respectively. Recall improved by 26.58% and 21.23%, while the precision of the LA view increased by 11.24%, and the precision of the AP view decreased by 12.61%. These results suggest that information transfer between views through the classification branch gradient is effective. Despite a decrease in precision for the AP views, the significant improvement in recall highlights the model’s enhanced ability to detect fracture areas.

Since the classification directly uses the information from two different views, the results are better than the detectors of both views, so we used the classification results to further find the fracture area. In this experiment, the detection results after the rescore strategy were validated. As shown in Table 9, the detector effects of the two views are also improved after the rescoring strategy, and the fracture detection IoU, accuracy, and recall of the AP views are increased by 0.0375, 0.59%, and 2.58%, respectively. For the LA view, IoU decreased by 0.0022 while the accuracy and recall increased by 3.94% and 12.85%.

Figure 9 shows the results of scaphoid fracture detection. In the figure, the red area represents the ground-truth box labeled by the doctor, while the green area shows the result predicted by the model. Figure 10 provides an example of a more obvious fracture, whereas Figure 11 illustrates an example of an occult fracture. These examples highlight the challenges in diagnosing scaphoid fractures, as occult fractures are difficult to distinguish by the naked eye. The AP views demonstrate a relatively high IoU, while the LA views also accurately locate the fracture area. This combined approach is effective in drawing the doctor’s attention to the area, enabling quicker identification of the fracture location.

## 4. Discussion

This article proposes a multi-view-based method for the classification and detection of scaphoid fractures. The results of correct fracture classification and the detection ratio reached 89.94% and 87.16%, respectively. The method has the potential to improve clinical radiologists’ diagnoses. Next, we address some limitations and ideas encountered during the experiments.

One challenge is the method used for labeling fractures. As mentioned earlier, fractures are confirmed by surgery, and labeled areas are identified using images from two different views simultaneously. This method may include fractured areas visible in only one view and not detectable in another. Despite this, the model learns the correspondence between views during training, enabling it to identify subtle differences between occult fractures and normal scaphoid bones. Consequently, the multi-view fusion model proposed in this study significantly improves the recall rate in both the fracture classification and detection branches.

Another issue is the quality of X-ray images. Although all images are from the same hospital, variations in shooting parameters and environmental conditions affect image quality. Figure 12 shows that images suffer from poor detail, contrast, or improper film darkness. Variations in the distance between the arm and the sensor led to differences in scaphoid bone slice size and resolution after resizing. Additionally, severe fractures, bone changes over time, and imaging artifacts from arthritis can complicate the training process. High-quality images are challenging to obtain because of these variances in patient conditions and imaging practices. Study [17] mentions eight conditions that lead to image rejection, such as unclear ground truth or contradictory radiologic diagnoses. In our dataset, images were annotated by an orthopedic surgeon but not reviewed by a radiologist, which may include images of patients or non-patients with similar characteristics, complicating the training process.

Figure 13 shows examples of prediction errors. The first row displays a false positive where a non-patient is mistakenly classified as having a fracture because of similar characteristics. The second row shows false negative cases that patients where the LA view resemble a normal scaphoid, resulting in no identifiable fracture area. Expanding the dataset could mitigate these issues.

Our study reveals that the classification branch outperforms the fracture detection branch in terms of performance. This is expected as the classification branch utilizes features from both views, while object detection is inherently more challenging. We aimed to enhance the detection branch using information from the classification branch by incorporating classification scores into the detection process. Despite these adjustments, a performance gap remains because classification scores lack spatial information crucial for detection tasks.

To address this, we experimented with the attention mechanism to extract important spatial features, multiplying them with classification scores to obtain spatial weights and enhance feature maps. Additionally, we added a prediction heat map branch to capture important spatial regions. However, these methods primarily accelerated model convergence during training and did not significantly improve performance in testing, likely because of overfitting from the small dataset.

## 5. Conclusions

In this article, we propose a high-performance automated scaphoid fracture diagnosis system that integrates information from both AP and LA views. The system provides an image-level classification to determine if the scaphoid is fractured and identifies the fracture location in both views. We enhance the detection network for both views by incorporating a multi-view classification branch that merges features from both views. During training, this fusion indirectly transfers information between views, improving the performance of both the classification branch and fracture detection results. This approach, particularly in terms of recall, allows the model to identify fracture locations more confidently by supplementing information from the other views. In the feature extraction stage, we introduce a context extractor with a broad receptive field. This module captures global information, enhancing the model’s understanding of the entire scaphoid bone and making it easier to detect subtle differences in similar environments. The experimental results show improved performance with this module. Additionally, we propose a multi-view feature fusion module akin to the Feature Pyramid Network (FPN). This module combines features from two views, aligning them according to their corresponding layers and adding them layer by layer. We use weight transform to balance the information between views and prevent information dilution. In the prediction stage, a rescoring strategy is applied by incorporating the classification score into the anchor box score. This approach aims to identify potential fracture areas more effectively, especially in regions that may indicate occult fractures. The automated scaphoid fracture detection system demonstrates strong performance, assisting doctors in diagnosing scaphoid fractures more efficiently. It not only classifies whether the scaphoid is fractured but also provides insights into the likely fracture areas from both views. Despite the good performance achieved in this study, information from different views is still passed indirectly. As discussed earlier, multi-view techniques in medicine are typically used for classification tasks or 3D reconstruction, not direct detection. While our approach significantly improves recall, it does not substantially enhance overall accuracy because of the indirect nature of information transfer.

## Figures and Tables

**Figure 1 diagnostics-14-02425-f001:**
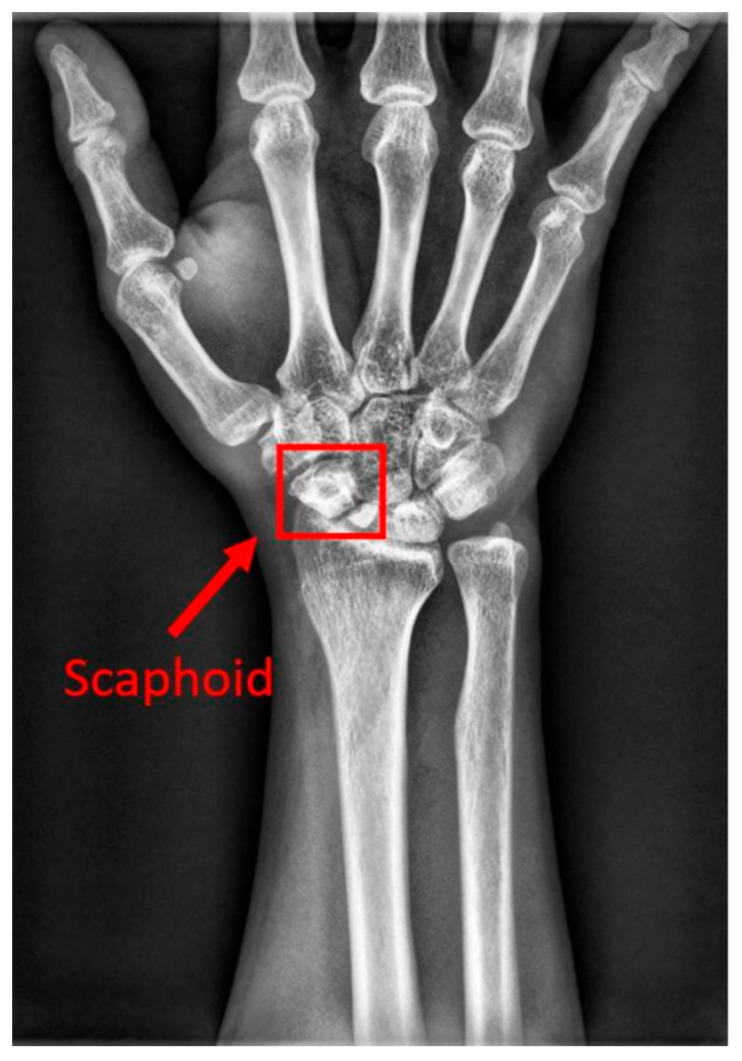
The location of the scaphoid bone. The scaphoid bone is within the red box.

**Figure 2 diagnostics-14-02425-f002:**
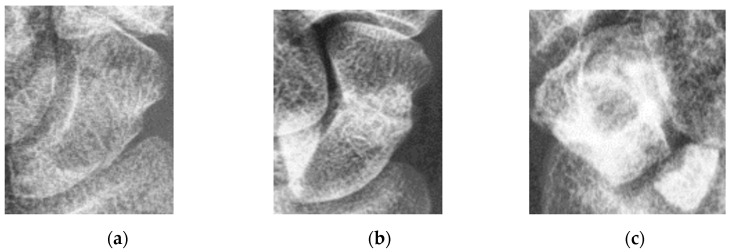
Scaphoid bone fractures under different conditions. (**a**) Occult fracture; (**b**) non-displaced fracture (hairline crack); and (**c**) displaced fracture.

**Figure 3 diagnostics-14-02425-f003:**
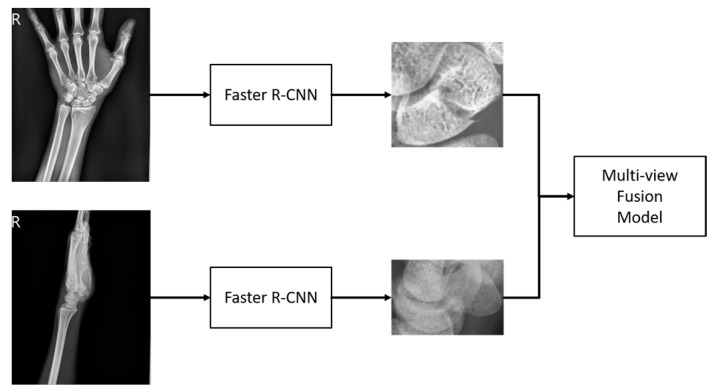
System overview.

**Figure 4 diagnostics-14-02425-f004:**
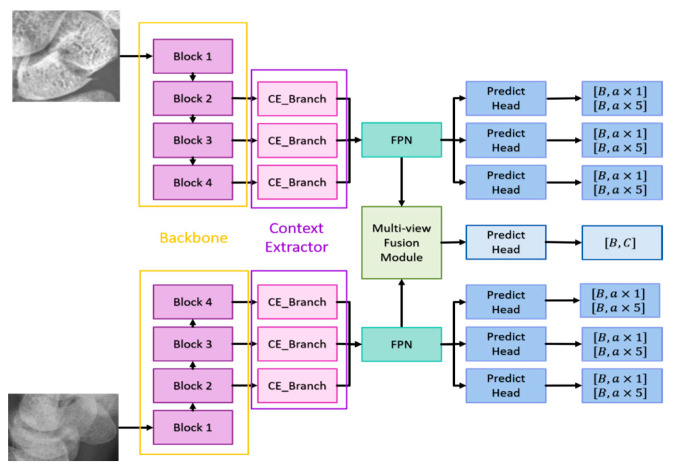
Overview of the multi-view fusion model.

**Figure 5 diagnostics-14-02425-f005:**
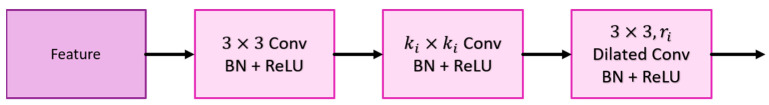
Context extraction.

**Figure 6 diagnostics-14-02425-f006:**
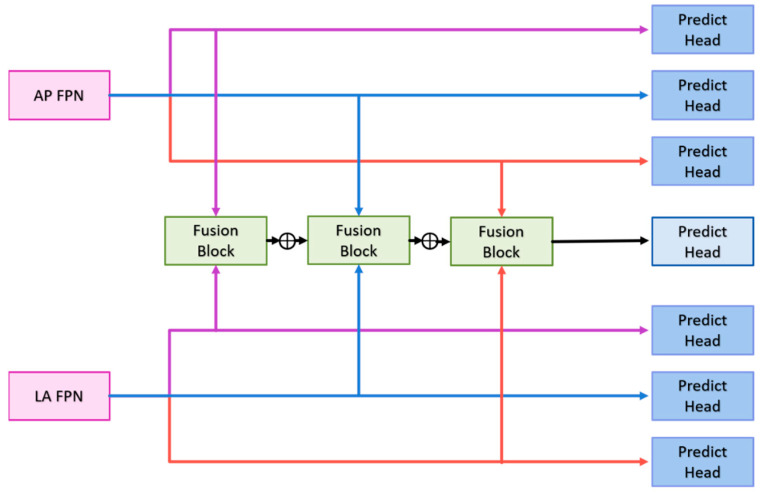
Multi-view fusion module architecture.

**Figure 7 diagnostics-14-02425-f007:**
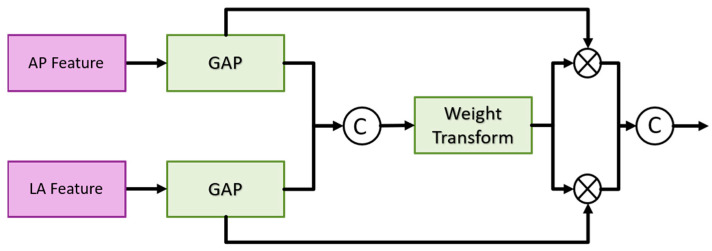
Fusion block architecture in the multi-view fusion module.

**Figure 8 diagnostics-14-02425-f008:**
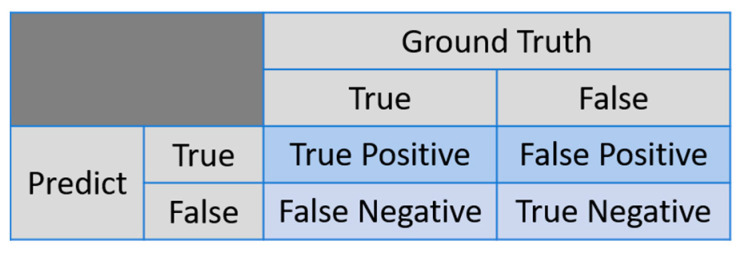
Confusion matrix.

**Figure 9 diagnostics-14-02425-f009:**
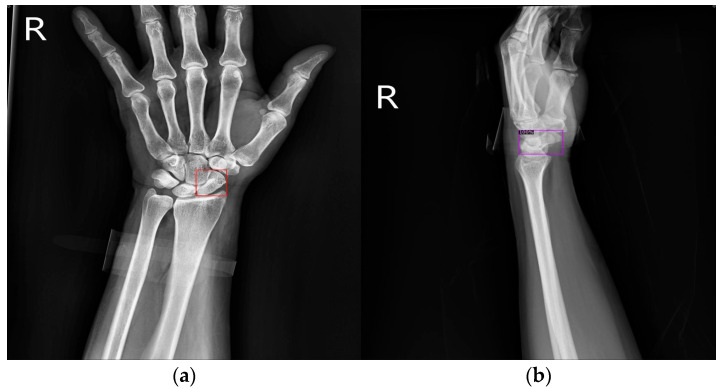
Scaphoid detection visualization. (**a**–**d**) The AP and LA views of the same patient. The first row is the ground truth, and the second row is the predicted result.

**Figure 10 diagnostics-14-02425-f010:**
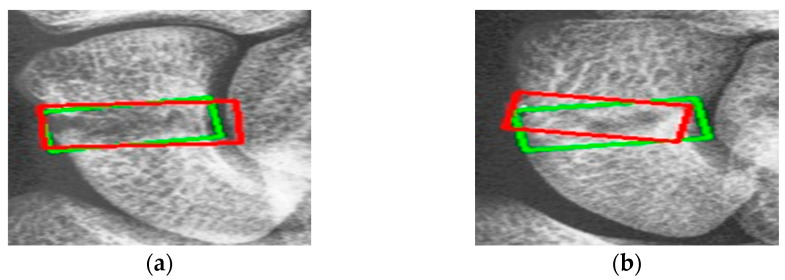
Fracture detection visualization. (**a**,**b**) Scaphoid bone slices. The red box is the true answer, and the green box is the predicted result.

**Figure 11 diagnostics-14-02425-f011:**
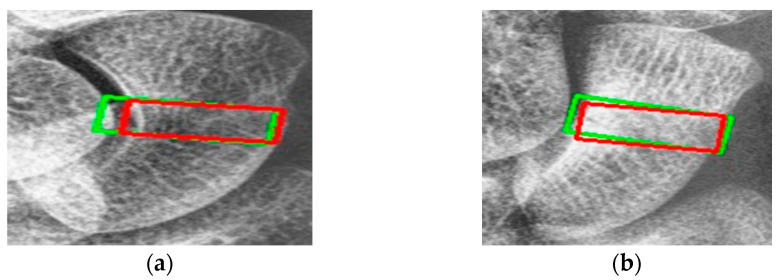
Occult fracture detection visualization. (**a**,**b**) Scaphoid bone slices. The red box is the true answer, and the green box is the predicted result.

**Figure 12 diagnostics-14-02425-f012:**
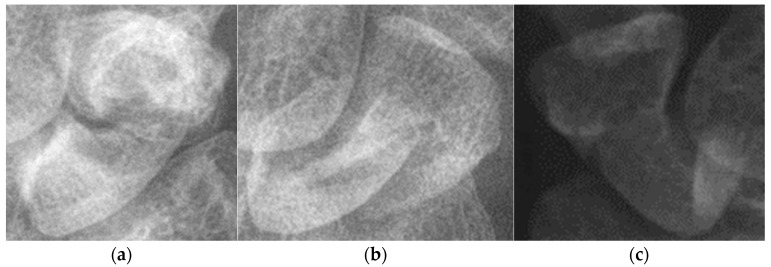
Poor quality images. (**a**) Light and shadow problems. (**b**) Too-severe displaced fractures. (**c**) Image with poor contrast.

**Figure 13 diagnostics-14-02425-f013:**
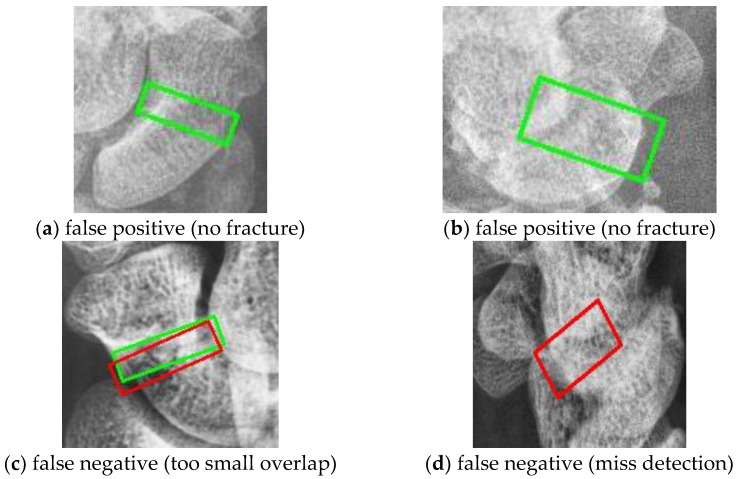
Examples of false positiveand false negative. The red box is fracture ground truth and green box is the predicted results.

**Table 1 diagnostics-14-02425-t001:** Experimental dataset.

Dataset	The Number of Normal	The Number of Fracture
Training dataset	80	60
Test dataset	20	15

**Table 2 diagnostics-14-02425-t002:** Results of the one-stage and two-stage detection methods for scaphoid detection.

Method	Recall	Precision
YOLO V4	97.50%	100.0%
Faster RCNN	100.0%	100.0%

**Table 3 diagnostics-14-02425-t003:** Comparison of single AP view and multi-view on classification branch.

Method	Accuracy	Recall	Precision	Sensitivity	Specificity
Single AP view	87.69%	75.26%	96.92%	75.26%	98.00%
Multi-view	87.65%	89.67%	84.67%	89.67%	86.00%

**Table 4 diagnostics-14-02425-t004:** Comparison of single-view and multi-view on detection branch.

Branch	Method	IoU	Accuracy	Recall	Precision
AP	Single AP view	0.5637	81.00%	58.25%	98.00%
	Multi-view	0.5509	85.44%	76.00%	90.85%
LA	Single LA view	0.4466	69.76%	56.06%	77.50%
	Multi-view	0.4697	74.32%	56.76%	80.65%

**Table 5 diagnostics-14-02425-t005:** Results of the context extractor on three branches in multi-view (X: not used; V: used).

Branch	CE	IoU	Accuracy	Recall	Precision
Classification	X		87.65%	89.36%	84.67%
	V		89.94%	87.33%	90.36%
AP Detection	X	0.5509	85.44%	76.00%	90.85%
	V	0.5109	86.57%	82.25%	87.45%
LA Detection	X	0.4697	74.32%	56.76%	80.65%
	V	0.4276	79.89%	64.44%	88.19%

**Table 6 diagnostics-14-02425-t006:** Scaphoid area detection results in AP and LA views obtained by using Faster RCNN.

Method	IoU	Accuracy	Recall	Precision
AP view	0.8662	100.0%	100.0%	100.0%
LA view	0.8478	100.0%	100.0%	100.0%

**Table 7 diagnostics-14-02425-t007:** Results of five-fold cross-validation of fracture classification on the single-view and multi-view fusion models (S: single AP view, M: multi-view).

Fold	Method	Accuracy	Recall	Precision	Sensitivity	Specificity
Fold 1	S	91.67%	81.25%	100.0%	81.25%	100.0%
	M	91.67%	87.50%	93.33%	87.50%	95.00%
Fold 2	S	91.67%	81.25%	100.0%	81.25%	100.0%
	M	94.44%	87.50%	100.0%	87.50%	100.0%
Fold 3	S	91.67%	84.21%	100.0%	84.21%	100.0%
	M	86.11%	93.75%	78.95%	93.75%	80.00%
Fold 4	S	80.56%	56.25%	100.0%	56.25%	100.0%
	M	88.89%	81.25%	92.86%	81.25%	95.00%
Fold 5	S	82.86%	73.33%	84.62%	73.33%	90.00%
	M	88.57%	86.67%	86.67%	86.67%	92.00%
Average	S	87.69%	75.26%	96.92%	75.26%	98.00%
	M	89.94%	87.33%	90.36%	87.33%	92.00%

**Table 8 diagnostics-14-02425-t008:** Results of five-fold cross-validation of fracture detection of AP view on the single-view and multi-view fusion models (S: single AP view, M: multi-view).

Fold	Method	IoU	Accuracy	Recall	Precision
Fold 1	S	0.6037	86.11%	68.75%	100.0%
	M	0.5608	88.89%	93.75%	83.33%
Fold2	S	0.5315	80.56%	56.25%	100.0%
	M	0.5657	88.89%	87.50%	87.50%
Fold 3	S	0.5231	77.78%	50.00%	100.0%
	M	0.4941	86.11%	87.50%	77.78%
Fold 4	S	0.6248	80.56%	56.25%	100.0%
	M	0.5994	83.33%	68.75%	91.67%
Fold 5	S	0.5353	80.00%	60.00%	90.00%
	M	0.5219	88.57%	86.67%	86.67%
Average	S	0.5637	81.00%	58.25%	98.00%
	M	0.5784	87.16%	84.83%	85.39%

**Table 9 diagnostics-14-02425-t009:** Results of the rescore strategy in two different views (V: used; X: no used).

Branch	Rescore	IoU	Accuracy	Recall	Precision
AP view	X	0.5109	86.57%	82.25%	87.45%
	V	0.5484	87.16%	84.83%	85.39%
LA view	X	0.4276	79.89%	64.44%	88.74%
	V	0.4254	83.83%	77.29%	88.19%

## Data Availability

The original contributions presented in the study are included in the article, further inquiries can be directed to the corresponding author.

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
