# Peer review of "The Detection and Classification of Scaphoid Fractures in Radiograph by Using a Convolutional Neural Network"

_diagnostics, 2024, doi:10.3390/diagnostics14212425_

Round 1
Reviewer 1 Report
Comments and Suggestions for Authors
1. Include references from more current studies CNN and online diagnostic tools. No articles in the references from 2023 or 2024 were present. Just three articles from 2022.
2. More details regarding the selection of specific parameters (such as dataset sizes and model architecture) could be beneficial to readers.
3. offer more thorough analyses of the findings, with a focus on how the single and multi-view techniques performed differently from one another.
4. Somewhere incorporate the limitation of the study. I think didn't found in it. Correct if already included
Author Response
Comments 1. Include references from more current studies CNN and online diagnostic tools. No articles in the references from 2023 or 2024 were present. Just three articles from 2022.
Response: The new references 31 and 31 are added in lines 117-125.
Comment 2. More details regarding the selection of specific parameters (such as dataset sizes and model architecture) could be beneficial to readers.
Response: The description of data sizes has been revised in lines153-157. The model trained method are explained in lines 402-417.
Commet 3. offer more thorough analyses of the findings, with a focus on how the single and multi-view techniques performed differently from one another.
Response: The performance analysis are shown in Tables 6, 7 and 8. The results demonstrated the fracture classification and detection of the multi-view CNN outperformed the AP view and LA view.
Comment 4. Somewhere incorporate the limitation of the study. I think didn't found in it. Correct if already included
Response: The limitation of the study had been explained in lines 546-589.
Reviewer 2 Report
Comments and Suggestions for Authors
This paper proposed an automated scaphoid fracture diagnosis system by integrating information from both AP and LA views. Before publishing, the author needs address these issues.
1. Lacking experimental analysis. Why Single AP view obtain better accuracy, precision, and specificity than multi-view in Table 2? Can you explain it?
2. Line 488. Faster RCnn à ‘Faster RCNN’.
3. Lacking ablation study. The effectiveness of each module is not discussed.
4. The clinical value still needs more clarification.
5. The limitation should be discussed.
Comments on the Quality of English LanguageN/A
Author Response
Comment 1. Lacking experimental analysis. Why Single AP view obtain better accuracy, precision, and specificity than multi-view in Table 2? Can you explain it?
Response: In our past study of reference 22, a CNN model was proposed to classify and detect the fracture of scaphoid. This method used the Global Average Pooling (GAP) to flatten and combine features from different layers via Feature Pyramid Networks (FPN). In order to compare the results and here proposed method, the experiments of Table 2 and Table 3 are implemented. The results are explained in lines 463-482.
Comment 2. Line 488. Faster RCnn à ‘Faster RCNN’.
Response: The mistakes had been modified in line 495.
Comment 3. Lacking ablation study. The effectiveness of each module is not discussed.
Response: The ablation study had been completed and further explained in lines 447-456. The detailed explanations are shown in Tables 6-8 and discussed in lines 501-527.
Comment 4. The clinical value still needs more clarification.
Response: The explanation is lines 546-548.
Comment 5. The limitation should be discussed.
Response: The limitation of the study had been explained in lines 546-589.
Round 2
Reviewer 2 Report
Comments and Suggestions for Authors
The author has addressed all my questions satisfactorily. This paper is now suitable for acceptance.
Comments on the Quality of English LanguageThe English in the paper can be improved before publication for clarity and fluency.